# Aperiodic approximants bridging quasicrystals and modulated structures

Toranosuke Matsubara [1], Akihisa Koga [1] ✉, Atsushi Takano[2], Yushu Matsushita [3] & Tomonari Dotera [4] ✉

Aperiodic crystals constitute a class of materials that includes incommensurate (IC) modulated structures and quasicrystals (QCs). Although these two categories share a common foundation in the concept of superspace, the relationship between them has remained enigmatic and largely unexplored. Here, we show "any metallic-mean" QCs, surpassing the confines of Penrose-like structures, and explore their connection with IC modulated structures. In contrast to periodic approximants of QCs, our work introduces the pivotal role of "aperiodic approximants", articulated through a series of $k$-th metallic-mean tilings serving as aperiodic approximants for the honeycomb crystal, while simultaneously redefining this tiling as a metallic-mean IC modulated structure, highlighting the intricate interplay between these crystallographic phenomena. We extend our findings to real-world applications, discovering these tiles in a terpolymer/homopolymer blend and applying our QC theory to a colloidal simulation displaying planar IC structures. In these structures, domain walls are viewed as essential components of a quasicrystal, introducing additional dimensions in superspace. Our research provides a fresh perspective on the intricate world of aperiodic crystals, shedding light on their broader implications for domain wall structures across various fields.

Prior to the discovery of quasicrystals (QCs) as the advent of aperiodicity in materials science, incommensurate (IC) modulated structures and IC composite structures were investigated, wherein IC spatial modulations were added to the background crystalline structures[1,2]. Then, the concept of superspace and additional degrees of freedom known as phasons were introduced. After Shechtman's discovery[3], aperiodic crystals, including IC modulated structures and QCs, emerged as an important class of materials[4–8]. Aperiodicity is characterized by irrational numbers, thereby making a distinction between the two. In QCs, the irrational numbers are locked by two-length scales[9–12] in geometry, whereas in IC modulated structures, these numbers remain unlocked. QCs typically consist of concentric shell clusters arranged quasiperiodically, locking in the golden mean in

icosahedral QCs. In certain alloys, such as Au-Al-Yb[13], periodic approximants are synthesized, where the clusters are arranged periodically. Consequently, "periodic approximants" have been extensively studied to gain a better understanding of QCs. A crucial aspect of periodic approximants is that they exhibit local quasiperiodicity (resembling QCs), but globally display periodicity. Moreover, as the degree of approximation increases, these periodic approximants converge towards QCs[14].

A complementary treatment has been explored where a quasiperiodic structure approaches the periodic one by varying the characteristic irrational. Such treatments are known as "aperiodic approximants"[15]. An elementary example of aperiodic approximants is the generalized Fibonacci sequence, which comprises two letters, $A$

[1]Department of Physics, Tokyo Institute of Technology, Meguro, Tokyo 152-8551, Japan. [2]Department of Molecular and Macromolecular Chemistry, Nagoya University, Nagoya, Aichi 464-8603, Japan. [3]Toyota Physical and Chemical Research Institute, Nagakute, Aichi 480-1192, Japan. [4]Department of Physics, Kindai University, Higashi-Osaka, Osaka 577-8502, Japan. ✉e-mail: koga@phys.titech.ac.jp; dotera@phys.kindai.ac.jp

and $B$. The sequence is generated by the substitution rules: $A \to AA \cdots AB (= A^k B)$ and $B \to A$, where $k$ is a natural number. The numbers of the letters $A$ and $B$ at iteration $n$ ($N_A^{(n)}$ and $N_B^{(n)}$) satisfy

$$\begin{pmatrix} N_A^{(n+1)} \\ N_B^{(n+1)} \end{pmatrix} = \begin{pmatrix} k & 1 \\ 1 & 0 \end{pmatrix} \begin{pmatrix} N_A^{(n)} \\ N_B^{(n)} \end{pmatrix}, \tag{1}$$

where the maximum eigenvalue of the matrix is given by the metallic-mean: $\tau_k = (k + \sqrt{k^2 + 4})/2$. When $k = 1$, the sequence is the conventional Fibonacci one with the golden mean. The eigenvector of the matrix is given by $(\tau_k, 1)^T$, indicating $N_A^{(n)}/N_B^{(n)} \to \tau_k$ as $n \to \infty$, where the sequence is filled with the letter $A$ for large values of $k$. In the limit $k \to \infty$, the sequence converges to a crystal consisting of consecutive "$A$"s. Hence, the generalized Fibonacci sequence with the metallic mean can be considered as the aperiodic approximants of the one-dimensional crystal $AAA \cdots$.

Similarly, aperiodic approximants of triangular lattices were proposed. These metallic-mean quasiperiodic tilings start from the bronze-mean tilings[11]. The majority of tiles increase with increasing $k$, and eventually, the systems converge to the triangular lattices in the limit $k \to \infty$. A crucial aspect of these is that they are locally periodic but globally quasiperiodic; in other words, they are considered planar IC-modulated structures.

Here, we present hexagonal metallic-mean approximants of the honeycomb lattice, which bridge the gap between QCs and IC modulated structures. A schematic of our view is presented in Fig. 1. As the metallic mean increases, the size of honeycomb domains bounded by the parallelograms also increases, and the whole tiling converges to the honeycomb lattice. Conversely, the metallic-mean IC modulation is introduced to the honeycomb crystals in terms of the metallic-mean tilings. The domain walls composed of parallelograms in the honeycomb crystal are regarded as ingredients of a quasicrystal adding superspace dimensions. Significantly, we show that the metallic-mean tiling scheme is applicable to a polymer system[16] and colloidal systems[17,18] in soft-matter self-assemblies.

## Results

### Metallic-mean tilings

We construct the metallic-mean approximants of the honeycomb lattice, which are composed of large hexagons (L), parallelograms (P), and small hexagons (S) shown in Fig. 2a. The ratio between the long ($\ell$) and short ($s$) lengths is given by the metallic-mean $\tau_k (= \ell/s)$. Consequently, the ratio of areas for the three tiles is given by $3\tau_k^2 : \tau_k : 3$. We elaborate the substitution rules for these tiles as a natural extension of those for the hexagonal golden-mean tiling[12] (Fig. 2b). The substitution rules for $k = 2$ and $k = 3$ are illustrated in Fig. 2c, d respectively. The details of the substitution rules are shown in Suppl. Note 1. Notice that the matching rule of the tilings is introduced by solid and open circles. When the deflation rule is applied to an L tile, an S tile is generated at the center of the original L tile, thereby, six zigzag chains of P tiles emanate from the central S tile, which is clearly found in the case with $k = 3$. The rest of the region is filled with L tiles. Upon one deflation process, a P tile is changed to one P tile and L tile, and an S tile is changed to one L tile. Hence, one can construct the substitution rules of three tiles for any $k$, which are subjected to the substitution rule for the generalized Fibonacci sequence: $\ell \to \ell^k s$ and $s \to \ell$. See also Suppl. Fig. 1 showing how these rules are extended to the cases of $k = 4$ and $k = 5$.

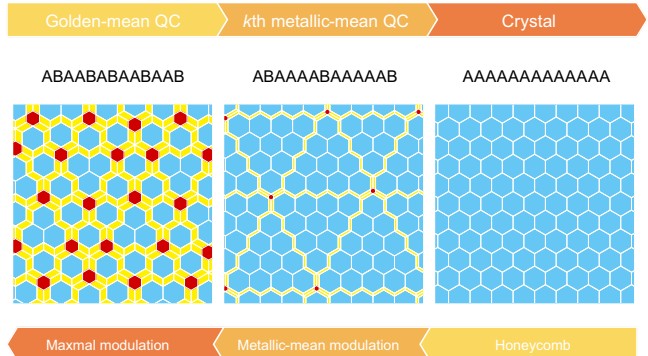

**Fig. 1 | Aperiodic approximants.** Schematic showing the role of aperiodic approximants as a link between quasicrystals and periodic crystals. The link is a series of $k$-th metallic-mean tilings as an aperiodic approximant of the honeycomb crystal (top arrow), which is simultaneously regarded as a metallic-mean (incommensurate) modulated honeycomb crystal (bottom arrow). Strings consisting of $A$ and $B$ represent the generalized Fibonacci sequences.

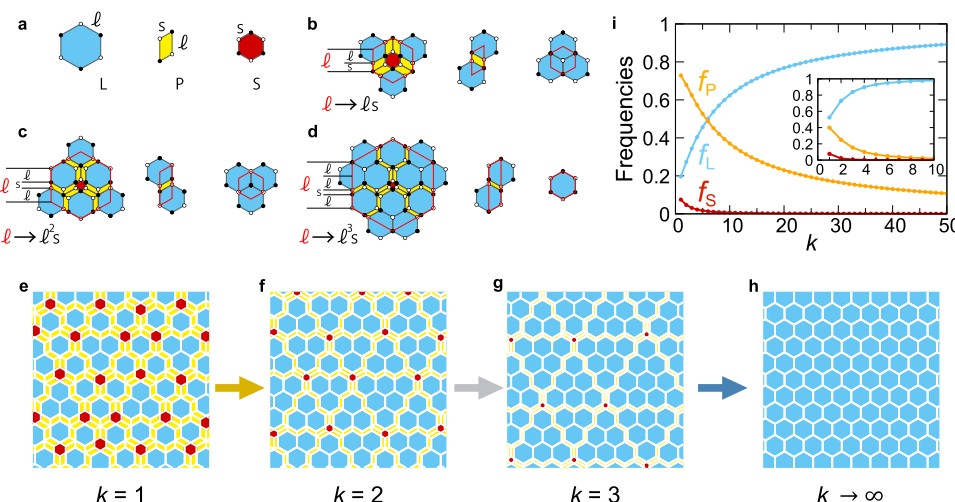

**Fig. 2 | Metallic-mean tilings. a** Large hexagons (L), parallelograms (P), and small hexagons (S) with edge lengths $\ell$ and $s$. Vertices are decorated with open and solid circles alternatively to introduce the matching rule of the tiling. **b–d** Substitution rules for the golden-mean tiling ($k = 1$)(**b**), silver-mean tiling ($k = 2$)(**c**), and bronze-mean tiling ($k = 3$)(**d**). **e** Golden-mean tiling. **f** Silver-mean tiling. **g** Bronze-mean tiling. **h** Honeycomb lattice. **i** Frequencies of L (blue), P (red), and S tiles (orange) as a function of $k$. Inset with the same axis labels shows the corresponding frequency for each area. The convergence to the periodic honeycomb lattice is assessed. Source data are provided as a Source Data file.

Two-dimensional space is covered without gaps after iterative deflation processes, as shown in Fig. 2e–g and Suppl. Fig. 2 for $k = 1 - 5$. Because of the deflation process, self-similarity is an inherent property of the metallic-mean tilings: Suppl. Fig. 3 exemplifies exact self-similarity for $k = 2$ and $k = 3$. We find that a finite number of adjacent L tiles are bounded by the P tiles, which can be regarded as an isolated "honeycomb domain". For example, in the case with $k = 2$, the domains are composed of one, three, or six L tiles, as shown in Fig. 2f. We confirm that each honeycomb domain bounded by the P tiles is composed of $a_{k-1}$, $a_k$, or $a_{k+1}$ adjacent L tiles in the $k$th metallic-mean tiling, where $a_k = k(k+1)/2$, see Suppl. Note 3. Therefore, by increasing $k$, the number of the L tiles in each honeycomb domain quadratically increases. On the other hand, the S and P tiles are located around the corners and edges of the honeycomb domains, and thereby their numbers should be $O(1)$ and $O(k)$, respectively. These suggest that the L tiles become the majority in the large $k$ case, and the single honeycomb domain is realized in the limit $k \to \infty$, as shown in Fig. 2h. Using a deflation matrix described in the Methods section, it is easy to evaluate the frequencies of tiles ($f_L$, $f_P$, and $f_S$) and the ratio of the corresponding areas ($S_L$, $S_P$, and $S_S$) rendered in Fig. 2i and its inset. For $k = 5$, more than ninety percent of the two-dimensional space is occupied by L tiles (see "Methods").

The tiling has eight types of vertices, as shown in Fig. 3a, classified by their coordination numbers and their circumstances. The frequency of each type can be exactly computed, and the explicit formulae for any $k$ are presented in Suppl. Note 2. Figure 3b shows the frequencies of the vertex types as a function of $k$. As expected, the frequencies of the $C_0$ and $C_1$ vertices monotonically increase and approach $1/2$, implying the convergence to the honeycomb lattice.

The vertices of the hexagonal metallic-mean tiling can be occupied alternatively by open and solid circles, corresponding to the A and B sublattices, respectively. This property is called bipartite. As shown in Fig. 3a, the vertex types $C_1$-$C_3$ belong to the A sublattice, and the others belong to the B sublattice, as depicted by open and solid circles, respectively. We find that the sublattice imbalance in the system given as $\Delta = f_A - f_B = 1/(3 + \sqrt{k^2 + 4})$, where $f_A (= f_{C_1} + f_{C_2} + f_{C_3})$ and $f_B (= f_{C_0} + f_{D_0} + f_{D_1} + f_E + f_F)$ are the frequencies of the A and B sublattices, respectively. This distinct property is in contrast to those for the bipartite Penrose, Ammann-Beenker, and Socolar dodecagonal tilings where each type of vertices equally belongs to both sublattices[19].

As shown in Fig. 3c, we can distinguish two kinds of L tiles denoted by $L_\triangle$ and $L_\triangledown$, introducing up and down triangles located at their centers so that three corners of each triangle point to the filled circles on the vertices of the L tile. In Fig. 3d, we find the following properties: (1) Two kinds of honeycomb domains composed of $L_\triangle$ or $L_\triangledown$ tiles are alternatively arranged. (2) The shape of $L_\triangle$ domains is up-triangular, and that of $L_\triangledown$ domains is down-triangular. (3) Domain walls are composed of consecutive zigzag P tiles. (4) Three domain walls should meet at a point of S hexagons. (5) Intervals between domain walls are not periodic, but metallic-mean incommensurately modulated (see "Methods" and Suppl. Fig. 6).

## Superspace representation

To provide a theoretical basis for the metallic-mean tilings, we construct their higher-dimensional description. In this method, the superspace is divided into the physical space and its complement, known as the perpendicular space. A tiling is viewed as a projection of a

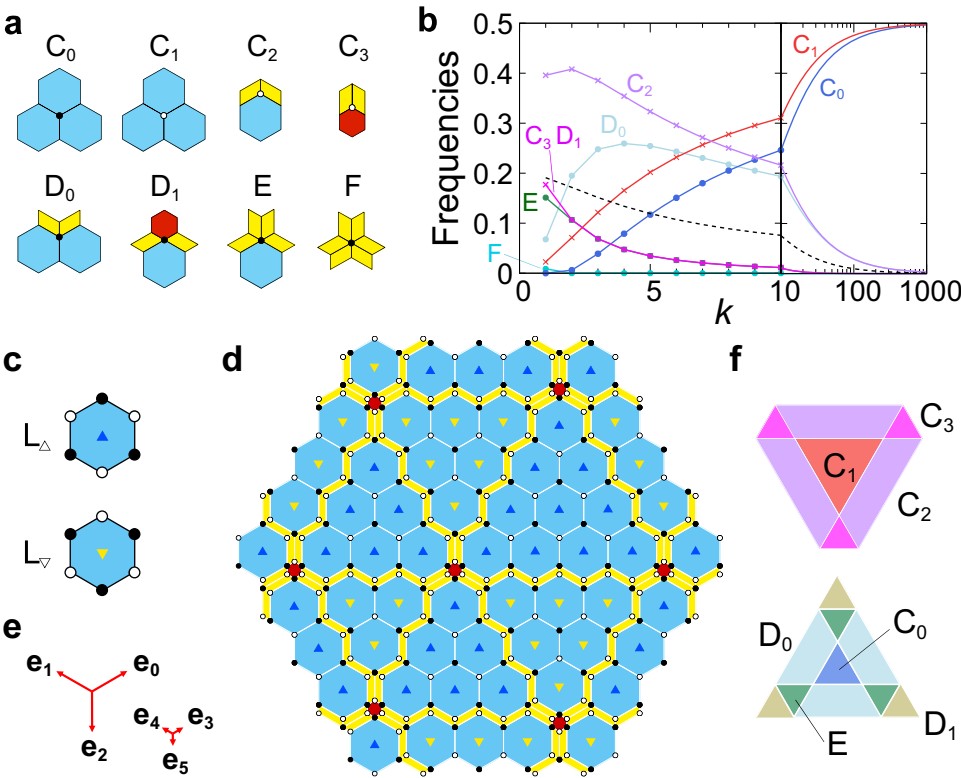

**Fig. 3 | Vertex types.** Vertices are alternatively decorated with open and solid circles to define A and B sublattices, respectively. **a** Eight types of vertices. **b** The blue, red, purple, light blue, green, and cyan lines represent the frequencies of $C_0$, $C_1$, $C_2$, $D_0$, E, and F vertices, respectively. The magenta line represents the frequencies of the $C_3$ and $D_1$ vertices. The dashed line represents the sublattice imbalance $\Delta$. **c** Two kinds of the L tiles, $L_\triangle$ and $L_\triangledown$. **d** Honeycomb domain structures for the hexagonal bronze-mean tilings. $L_\triangle$ tiles form up-triangular domains, and $L_\triangledown$ tiles form down-triangular domains. **e** Projected basis vectors $e_i$ ($i = 0, \cdots, 5$) from fundamental translation vectors in six dimensions. **f** Windows in the perpendicular space. Each area shows the vertex types. Source data are provided as a Source Data file.

hypercubic crystal in the superspace onto the two-dimensional physical space. The projections onto the perpendicular space are densely filled in specific areas, as illustrated in Fig. 3f, which are referred to as windows. These windows are derived from sections perpendicular to the threefold axis of a rhombohedron (octahedron), which is the projection of the hypercubic unit cell, showcasing hexagonal and triangular shapes (see details in "Methods"). The figure also highlights the regions associated with the eight vertex types, as detailed in the Methods section and Suppl. Fig. 8.

## Application to soft matter

The metallic-mean tilings are physical entities in two soft-matter systems. We consider self-assembled crystalline structures obtained in soft materials with the P31m plane group, as illustrated in Fig. 4a, which belongs to the two-dimensional hexagonal Bravais lattice but lacks hexagonal rotational axes. Further crystallographic description is given in Suppl. Notes 8 and 9 for colloidal particles and polymer blends, respectively.

The first application of the metallic-mean tiling is a polymer system reported by Izumi et al., who found a complex ordered structure in an ABC triblock terpolymer/homopolymer blend system[16] (for sample preparation, see Methods). Figure 4b illustrates the decoration of L, P, and S tiles by three kinds of polymers. In the previous study, regular large domains consisting of only L tiles were observed. It is noticed that the triangle inside a hexagon has two directions, up and down. In the present study, we searched samples again and found P tiles in a transmission electron microscopy (TEM) picture rendered in Fig. 4c. In Fig. 4c, a regular region of an extended L▽ area in the center and a domain wall represented by a row of zigzag P tiles on the left-hand side. We can interpret the rows of P tiles within the L sea as twin boundaries, which mark a transition between different crystal orientations, L△ and L▽. It's worth emphasizing that a row of P tiles physically changes the crystal

orientations, demonstrating the tangible properties of P tiles beyond mathematical concepts. We note that the decoration of S tile (Fig. 4b) is hypothetical and it has not been observed in the samples.

The second application of the metallic-mean tiling is a colloidal particle simulation in two dimensions conducted by Engel[17]. It utilizes a Lennard-Jones-Gauss (LJG) potential[20] that has two distinct length scales. We have reproduced his result and find that the LJG particles occupy the same positions as the dark gray circles in Fig. 4a–c. Moreover, in Fig. 2 of the Engel's paper and his Suppl. Fig. S1 in particular, it was shown to form twin-boundary superstructures on a scale much larger than the potential range: the size of superstructures depends on the temperature reversibly; the lower the temperature, the larger the size. One finds that regular L△ or L▽ domains form triangle shapes of several sizes, which property is also characteristic of the metallic-mean tiling. In addition, it was observed that twin boundaries only intersect at triple junctions, which situation mimics the metallic-mean tilings, where triple rows consisting of the P tiles meet at the location of an S hexagon, though the correspondence between the P tiles and domain walls in the LJG system is not always exact. Nonetheless, the metallic-mean scheme mimics Engel's modulated superstructures with changing scale ratios or $k$ values.

In Fig. 4d, an ideal decoration model for the particle system is generated by the higher-dimensional quasicrystal theory with the bronze-mean modulation (Suppl. Note 12). In these cases, as shown in Fig. 4e, f, the structure factor $S(\mathbf{q}) = |\frac{1}{N}\sum_i e^{i\mathbf{q}\cdot\mathbf{r}_i}|^2$ theoretically calculated in terms of the superspace representation dramatically reproduces the numerical Fourier transformations for the diffraction images shown in Fig. 3 of Engel's paper. As clearly shown in the magnified views (Fig. 4f), the prominent peaks appear at almost the same positions, while the aperiodic modulation of the metallic-mean tiling yields the satellite peaks in the vicinity of the main peaks, which is the characteristic property of IC structures.

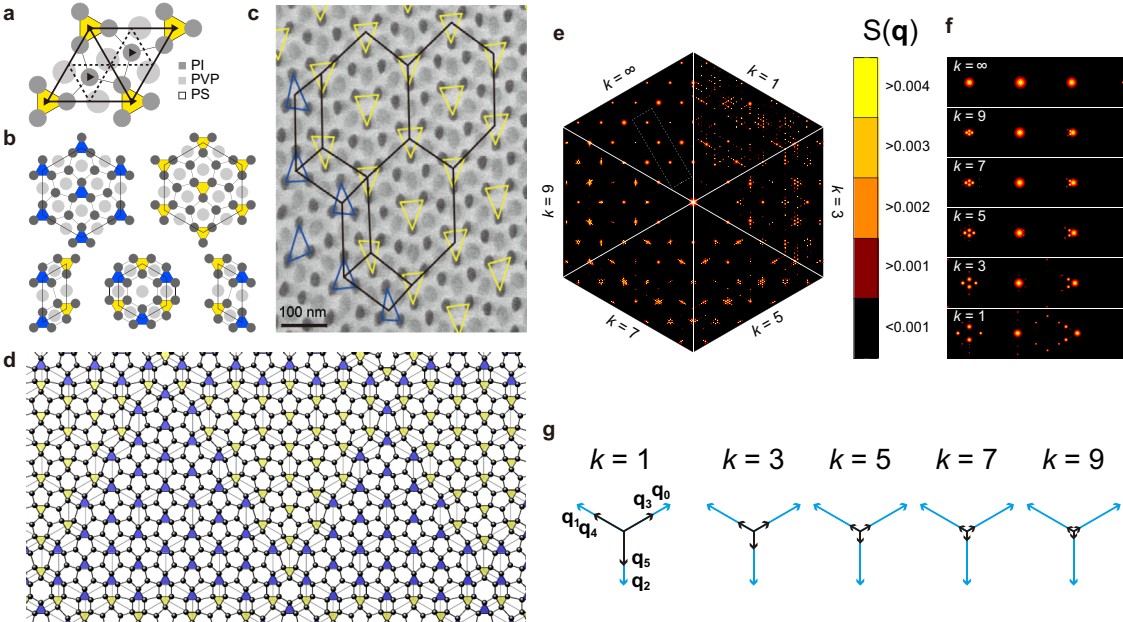

**Fig. 4 | Application to soft matter. a** Diagram of the P31m plane group. **b** Schematic decoration of L, P, and S tiles by ABC triblock terpolymer/homopolymer blend ISP-III/S. Dark gray circles indicate polyisoprene (PI), light gray circles indicate poly(2-vinylpyridine) (PVP), and the other matrix region is polystyrene (PS). **c** TEM image from the ABC triblock terpolymer/homopolymer ISP-III/S. **d** Ideal particle decoration for a colloidal system generated by the bronze-mean tiling. Up and down triangles form blue and yellow triangular domains reproducing colloidal

simulations. **e** Each sector shows the structure factor for the decorated $k$-th metallic-mean tiling when $k = 1, 3, 5, 7, 9$, and $\infty$. **f** Magnified views of slices of the structure factor indicated by a dashed rectangle in (**e**). In the vicinity of main peaks, the aperiodic modulation yields satellite peaks characterizing IC structures. **g** Six reciprocal vectors $\mathbf{q}_i$ ($i = 0, 1, \cdots, 5$) for $k = 1, 3, 5, 7$, and 9. Source data are provided as a Source Data file.

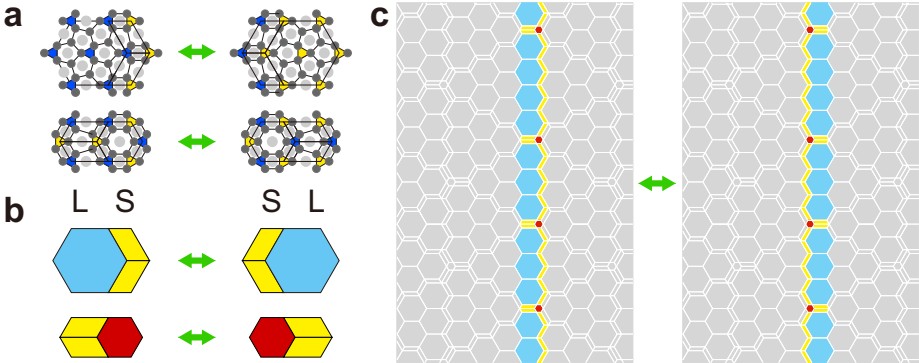

**Fig. 5 | Phason flips. a** Schematic phason moves in a self-assembled pattern from ABC triblock terpolymer/homopolymer blend. Colored circles and triangles represent the same as in Fig. 4a, b. **b** Two types of phason flips in the metallic-mean tiling. **c** Move of a twin boundary by a row of phason flips. The difference is explicitly shown in color.

## Discussion

Our previous study has covered the multiples-of-3 metallic-means, through the hexagonal aperiodic approximants of the triangular lattice[15]. The present work broadens the scope of aperiodic approximants. Firstly, our tiling serves as the approximant of the honeycomb lattice. Secondly, it enables an inflation ratio of any metallic mean, thereby enhancing the applicability.

In fact, we have applied the tiling concept to explore real materials, such as polymer and colloidal systems. Our analysis identifies large hexagons as regular structures and parallelograms as twin boundaries. It is noted that similar IC triangular domain structures were discovered in quartz and aluminum phosphate a long time ago[21,22], known as Dauphiné twins in trigonal quartz. We surmise that there is a similar mechanism behind the formation.

We emphasize that the decorated perpendicular space windows in 6D generate the 2D IC structures, whose method has been developed in the field of QC studies. It is important that the satellite peaks can be calculated not by direct real-space Fourier transform but by perpendicular-space Fourier transform of the windows. By comparing these peaks with those observed in two-dimensional IC modulated structures, we establish a foundation for the analysis of IC structures in terms of the QC methodology.

One of the origins of the P31m plane group demonstrated here is the aggregation tendency of pentagons. Regular pentagons cannot tile the entire plane without gaps, as shown by Dürer-Kepler-Penrose, however, there are pentagon-related tilings if we abort five-fold symmetry. In Suppl. Note 10, we demonstrate the accommodation of pentagons within both a square and a hexagon. Using 4-fold symmetry, the Cairo pentagonal tiling and its dual, i.e., the $3^2$. 4.3. 4 Archimedean tiling with the P4gm has been considered[23]. The latter Archimedean tiling is associated with the $\sigma$ phase found in complex metallic and soft-matter phases, which is recognized as a periodic approximant of dodecagonal QCs[24–29]. It is noteworthy that the P31m plane group structure is a 3-fold variant of the Cairo tiling and the $\sigma$ phase.

Our study highlights the effectiveness of aperiodic approximants in inducing modulations within self-assembled soft-matter systems employing the P31m plane group. Specifically, we utilized the rows of P tiles as domain boundaries in the honeycomb lattice, thereby bridging metallic-mean hexagonal QCs and IC modulated honeycomb lattices. The dynamic movement of domain walls while maintaining triple junctions can be explained by the phason flips of L, S, and P tiles, as illustrated in Fig. 5 and Suppl. Note 4. In this context, the colloidal system appears to be a phason-random tiling version of the metallic-mean tiling system. Lastly, applying the deterministic growth rules, known as Onoda-Steinhardt-DiVincenzo-Socolar (OSDS) rules[30], reveals that dead surfaces consist of these domain walls. Overall, our

research provides insights into the realm of both aperiodic crystals and their broader implications for domain wall structures across various fields.

## Methods

### Deflation matrix of the metallic-mean tiling

The metallic-mean tilings are regarded as the aperiodic approximants of the honeycomb lattice. To discuss quantitatively how the metallic-mean tilings approach the honeycomb lattice with increasing $k$, we construct the deflation matrix. At each deflation process, the increase of the numbers of L, P, and S tiles is explicitly given by $\mathbf{v}_{n+1} = M\mathbf{v}_n$ with $\mathbf{v}_n = (N_L^{(n)}, N_P^{(n)}, N_S^{(n)})^T$ and

$$
M = \begin{pmatrix} k^2 & \frac{k}{3} & 1 \\ 6k & 1 & 0 \\ 1 & 0 & 0 \end{pmatrix}, \tag{2}
$$

where $N_\alpha^{(n)}$ is the number of the tile $\alpha$, which stands for L, P, or S at iteration $n$. The maximum eigenvalue of the matrix $M$ is $\tau_k^2$, and the corresponding eigenvector is given as $(\tau_k^2, 6\tau_k, 1)^T$. We evaluate the frequencies for these tiles in the large $k$ limit approach $f_L = \tau_k^2/Z$, $f_P = 6\tau_k/Z$, $f_S = 1/Z$, where $Z = \tau_k^2 + 6\tau_k + 1$. The $k$-dependent frequencies for three tiles are shown in Fig. 2i. Increasing $k$, the frequency of the L tiles monotonically increases and approaches unity.

### Domain boundaries

The domain boundaries composed of consecutive zigzag P tiles intersect at small hexagons and pass through the opposite edge of the small hexagons while keeping alternating directions of P tiles. If we ignore these slithering configurations of P tiles, there are three sets of parallel domain walls, as displayed in Fig. 3d. Focusing on a set of parallel domain walls, we observe two types of intervals between the domain walls denoted by $\mathcal{S}_S$ and $\mathcal{S}_L$, as shown in Suppl. Fig. 6 for silver- and bronze-mean tilings. There are intriguing properties for the intervals. First, for the $k$-th metallic-mean tilings, the interval $\mathcal{S}_S$ and $\mathcal{S}_L$ consists of $k$ and $k+1$ consecutive L tiles. Second, upon the deflation, we find the substitution rules: $\mathcal{S}_L \to \mathcal{S}_L^{\frac{1}{2}} \mathcal{S}_S^k \mathcal{S}_L^{\frac{1}{2}}$, and $\mathcal{S}_S \to \mathcal{S}_L^{\frac{1}{2}} \mathcal{S}_S^{k-1} \mathcal{S}_L^{\frac{1}{2}}$. The numbers of intervals $N_{\mathcal{S}_S}^{(n)}$ and $N_{\mathcal{S}_L}^{(n)}$ of the $n$-th generation satisfy

$$
\begin{pmatrix} N_{\mathcal{S}_S}^{(n+1)} \\ N_{\mathcal{S}_L}^{(n+1)} \end{pmatrix} = \begin{pmatrix} k-1 & k \\ 1 & 1 \end{pmatrix} \begin{pmatrix} N_{\mathcal{S}_S}^{(n)} \\ N_{\mathcal{S}_L}^{(n)} \end{pmatrix}, \tag{3}
$$

where the maximum eigenvalue of the matrix is given by the metallic-mean $\tau_k$. The eigenvector of the matrix is given by

# a

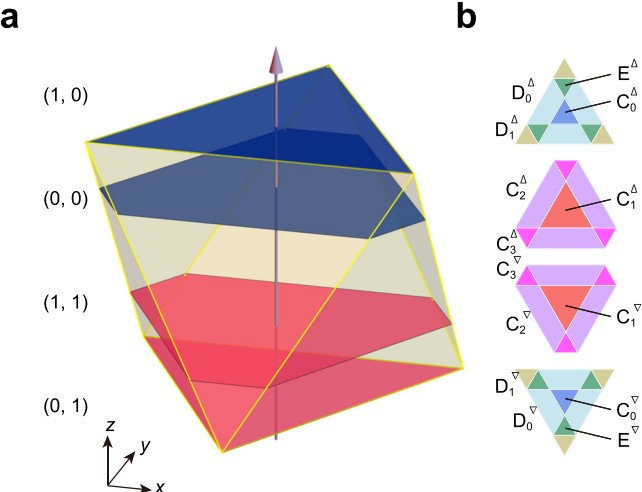

(1, 0)

(0, 0)

(1, 1)

(0, 1)

# b

$E^\triangle$
$D_0^\triangle$
$C_0^\triangle$
$D_1^\triangle$

$C_2^\triangle$
$C_1^\triangle$
$C_3^\triangle$
$C_3^\triangledown$

$C_2^\triangledown$
$C_1^\triangledown$

$D_1^\triangledown$
$C_0^\triangledown$
$D_0^\triangledown$
$E^\triangledown$

**Fig. 6 | Superspace perspective. a** the perpendicular space for the bronze-mean tiling. **b** Four windows on the right-hand side are obtained from a regular octahedron (middle part of a rhombohedron) of edge length $\sqrt{3}(1 + \tau_k^{-1})$. The top (1, 0) and bottom (0, 1) windows are equilateral triangular faces of the solid, and hexagonal windows indicated by (0, 0) and (1, 1) are the sections of the octahedron. In the solid, blue and red colors correspond to honeycomb domains with $L_\triangle$ and $L_\triangledown$, respectively. In each window, each color corresponds to the vertex type rendered in Fig. 3b.

$(\tau_k - 1/\tau_k, 1 + 1/\tau_k)^T$, indicating $N_{\mathcal{S}_S}^{(n)}/N_{\mathcal{S}_L}^{(n)} \to \tau_k$ as $n \to \infty$, where the sequence is filled with $\mathcal{S}_S$ intervals for large $k$ values. Therefore, we conclude that the intervals between domain walls are metallic-mean modulated.

## Superspace representation

We outline the main steps of the construction of the metallic-mean tiling by the projection of a higher-dimensional hyperlattice onto the physical space. Let $\ell$ and $s$ be the lengths of the long and short edges of the tiling. We here assume that the ratio $\eta = s/\ell$ is a variable to apply the tiling to soft-matter systems, while the ratio in the perpendicular space is set to be $1/\tau_k$ to keep the arrangement of the metallic-mean tiling. When $\eta = 1/\tau_k$, the tiling is the exact self-similar metallic-mean tiling generated by the deflation rules.

Each vertex site in the tiling is described by a six-dimensional lattice point $\vec{n} = (n_0, n_1, \cdots, n_5)^T$, labeled with integers $n_m$. Let the six-dimensional lattice point $\vec{r}^h$ in the six-dimensional space $\mathcal{S}^h$ as $\vec{r}^h = R\vec{n}$:

$$
R = \begin{pmatrix}
\ell c_6 & -\ell c_6 & 0 & s c_6 & -s c_6 & 0 \\
\ell s_6 & \ell s_6 & -\ell & s s_6 & s s_6 & -s \\
\tau_k^{-1} c_6 & -\tau_k^{-1} c_6 & 0 & -c_6 & c_6 & 0 \\
\tau_k^{-1} s_6 & \tau_k^{-1} s_6 & -\tau_k^{-1} & -s_6 & -s_6 & 1 \\
\sqrt{2}\tau_k^{-1} & \sqrt{2}\tau_k^{-1} & \sqrt{2}\tau_k^{-1} & 0 & 0 & 0 \\
0 & 0 & 0 & -\sqrt{2} & -\sqrt{2} & -\sqrt{2}
\end{pmatrix}, \quad (4)
$$

where $R$ is the mapping matrix and $c_6 = \cos(\pi/6)$, $s_6 = \sin(\pi/6)$. Namely, the matrix is represented by the six-dimensional basis vectors $\vec{e}_i^h$ ($i = 0, 1, \cdots, 5$): $(\vec{e}_i^h)_j = R_{ji}$. The vertex site **r** in the physical space $\mathcal{S}$ is given by the first two components of the vector: $\mathbf{r} = ((\vec{r}^h)_0, (\vec{r}^h)_1) = \sum_{m=0}^5 n_m \mathbf{e}_m$, where the projected vectors of the form $\mathbf{e}_m = (R_{0m}, R_{1m})$ with lengths $\ell$ and $s$ are displayed in Fig. 3e. The remaining perpendicular space is split into two-dimensional spaces $\tilde{\mathcal{S}}$ and $\mathcal{S}^\perp$, and the corresponding coordinates $\tilde{\mathbf{r}}$ and $\mathbf{r}^\perp$ are given as $\tilde{\mathbf{r}} = ((\vec{r}^h)_2, (\vec{r}^h)_3) = \sum_{m=0}^5 n_m \tilde{\mathbf{e}}_m$, $\mathbf{r}^\perp = ((\vec{r}^h)_4, (\vec{r}^h)_5) = \sum_{m=0}^5 n_m \mathbf{e}_m^\perp$, where $\tilde{\mathbf{e}}_m = (R_{2m}, R_{3m})$ and $\mathbf{e}_m^\perp = (R_{4m}, R_{5m})$.

Note that $\tilde{\mathbf{r}}$ points are densely filled on four planes with $\mathbf{r}^\perp = \{(\sqrt{2}\tau_k^{-1}, 0), (0,0), (\sqrt{2}\tau_k^{-1}, -\sqrt{2}), (0, -\sqrt{2})\}$ denoted by $\{(1,0), (0,0), (1,1), (0,1)\}$, having polygonal windows shown in Fig. 6. Notice that the windows are faces and sections for a regular octahedron. This octahedron is the middle part of a rhombohedron of edge length $\sqrt{3}(1 + \tau_k^{-1})$, which is the projection of the hypercubic unit cell. In Fig. 6, $\tilde{\mathbf{r}}$ is plotted in the $(x, y)$-directions, while for $\mathbf{r}^\perp$ both $(\vec{r}^h)_4$ and $(\vec{r}^h)_5$ are projected onto the $z$ component. We find that in the limit $k \to \infty$, the upper and lower hexagons get closer to the top and bottom faces, respectively, and finally, they become equilateral triangles. The explicit sizes of hexagonal windows are presented in Suppl. Fig. 8.

The six-dimensional reciprocal lattice vectors $\vec{q}_i^h$ are defined to have the following property $\vec{e}_i^h \cdot \vec{q}_j^h = 2\pi \delta_{ij}$ with $\delta_{ij}$ is the Kronecker delta. It is easy to find $(\vec{q}_j^h)_i = Q_{ij}$, where $RQ^T = 2\pi \delta_{ij}$ and

$$
Q = C \begin{pmatrix}
c_6 & -c_6 & 0 & \tau_k^{-1} c_6 & -\tau_k^{-1} c_6 & 0 \\
s_6 & s_6 & -1 & \tau_k^{-1} s_6 & \tau_k^{-1} s_6 & -\tau_k^{-1} \\
s c_6 & -s c_6 & 0 & -\ell c_6 & \ell c_6 & 0 \\
s s_6 & s s_6 & -s & -\ell s_6 & -\ell s_6 & \ell \\
\psi & \psi & \psi & 0 & 0 & 0 \\
0 & 0 & 0 & -\psi \tau_k^{-1} & -\psi \tau_k^{-1} & -\psi \tau_k^{-1}
\end{pmatrix} \quad (5)
$$

with $C = 4\pi/[3(\ell + s \tau_k^{-1})]$ and $\psi = (\ell \tau_k + s)/(2\sqrt{2})$. $\mathbf{q} = ((\vec{q}_i^h)_0, (\vec{q}_i^h)_1) = \sum_{m=0}^5 n_m \mathbf{q}_m$, where the projected vectors $\mathbf{q}_m = (Q_{0m}, Q_{1m})$ with lengths $1/\ell$ and $1/(\ell \tau_k)$ are displayed in Fig. 4g. The remaining four-dimensional perpendicular space is split into two-dimensional reciprocal spaces and the corresponding reciprocal vectors $\tilde{\mathbf{q}}$ and $\mathbf{q}^\perp$ are given as $\tilde{\mathbf{q}} = ((\vec{q}_i^h)_2, (\vec{q}_i^h)_3) = \sum_{m=0}^5 n_m \tilde{\mathbf{q}}_m$, $\mathbf{q}^\perp = ((\vec{q}_i^h)_4, (\vec{q}_i^h)_5) = \sum_{m=0}^5 n_m \mathbf{q}_m^\perp$, where $\tilde{\mathbf{q}}_m = (Q_{2m}, Q_{3m})$ and $\mathbf{q}_m^\perp = (Q_{4m}, Q_{5m})$. The detailed procedure is given in Suppl. Note 5.

When computing the fast Fourier transforms (FFT), we rely on the following identity for any pair of vectors in the superspace lattice $\vec{r}^h$ and in the corresponding reciprocal lattice $\vec{q}^h$: $1 = \exp(i\vec{q}^h \cdot \vec{r}^h) = \exp(i\mathbf{q} \cdot \mathbf{x}) \exp(i\tilde{\mathbf{q}} \cdot \tilde{\mathbf{x}}) \exp(i\mathbf{q}^\perp \cdot \mathbf{x}^\perp)$. If particle's positions are described by $\delta$-functions so that the density reads $f(\mathbf{r}) = \sum_{j=1}^N \delta(\mathbf{r} - \mathbf{r}_j)$, then the Fourier transform of the density is calculated as

$$
\int d\mathbf{r}\, e^{-i\mathbf{q}\cdot\mathbf{r}} f(\mathbf{r}) = \sum_{j=1}^N e^{-i\mathbf{q}\cdot\mathbf{x}_j} = \sum_{j=1}^N e^{i\tilde{\mathbf{q}}\cdot\tilde{\mathbf{x}}_j} e^{i\mathbf{q}^\perp \cdot \mathbf{x}_j^\perp}, \quad (6)
$$

in the last step, we resorted to the above identity.

To construct decorated tilings (Fig. 4d) for soft-matter systems, we set $\eta = s/\ell = 0.6249$. In this case, we employ sections of a rhombohedron as extensional windows. Detailed procedures are presented in Suppl. Notes 8 and 12.

## Polymer details

An ISP (I: polyisoprene, S: polystyrene, P: poly(2-vinylpyridine)) triblock terpolymer sample was prepared by a sequential monomer addition technique of an anionic polymerization from cumylpotassium as an initiator in tetrahydrofuran (THF), while styrene homopolymer was synthesized anionically with *sec*-butyllithium in benzene. The average molecular weight of the terpolymer is 161 kg mol⁻¹ and the composition is $\phi_I/\phi_S/\phi_P = 0.25/0.53/0.22$, whereas that of the styrene homopolymer is 9 kg mol⁻¹. The overall composition of the blend sample is $\phi_I/\phi_S/\phi_P = 0.17/0.68/0.15$, where polystyrene block/styrene homopolymer ratio of $w_S(b)/w_S(h) = 1.4$. The sample film was obtained by casting for two weeks from a dilute solution of THF, followed by heating at 150 °C for two days. The specimens for morphological observation were cut by an ultramicrotome

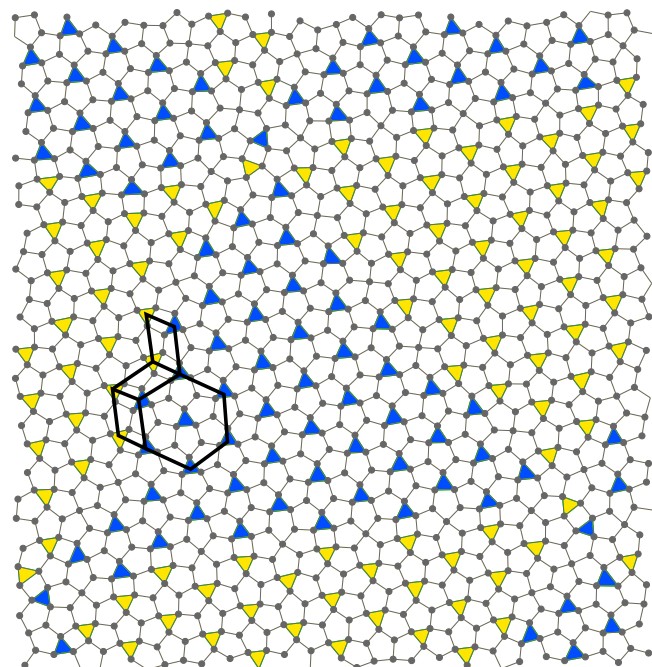

**Fig. 7 | Colloidal simulation.** Monte Carlo simulation for the Lennard-Jones-Gauss potential at $T = 0.270$. Different colored triangles represent different honeycomb domains.

of Leica model Ultracut UCT into ultrathin sections of about 100 nm thickness and stained with $OsO_4$ for the TEM observation. Further details are provided in ref. 16.

### Simulations of colloidal particles

We used *NPT* (constant number of particles *N*, external pressure *P*, and temperature *T*) Monte Carlo simulations of $N = 10{,}000$ colloidal particles interacting with the Lennard-Jones-Gauss potential[17,20] given by

$$V(r) = \frac{1}{r^{12}} - \frac{2}{r^6} - \epsilon \exp\left(-\frac{(r - r_0)^2}{2\sigma^2}\right), \qquad (7)$$

with parameters $\sigma^2 = 0.042$, $\epsilon = 1.8$, $r_0 = 1.42$ at $T = 0.270$, $P = 0.0$. There are slight differences between simulations (Fig. 7) and the metallic-mean tiling model (Fig. 4): (1) Dynamically, P tiles are not always perfect. (2) There are five particles in an S tile in simulations, while six particles in the latter. The effect of these is negligible in the structure factor. Further data including diffraction images is provided in Suppl. Note 11.

### Phasons

Domain walls dynamically move with keeping triple junctions can be explained by the phason flips of L, S, and P tiles, as shown in Fig. 5. In this sense, the colloidal system appears to be a phason-random tiling version of the metallic-mean tiling system. The existence and the conservation of S tiles in the phason flips is the key of triple junctions of domain walls at moderate thermal excitations (see also Suppl. Note 4).

## Data availability

The data that support the findings of this study are available from the corresponding author upon request. Source data are provided with this paper.

## Code availability

The codes used to construct the tilings and the projection windows are available from the corresponding author upon request.

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

## Acknowledgements
The authors thank M. Engel, M. Mihalkovic, and P. Ziherl for valuable discussions. Parts of the numerical calculations are performed in the supercomputing systems in ISSP, the University of Tokyo. This work was supported by Grant-in-Aid for Scientific Research from JSPS, KAKENHI Grant Nos. JP22K03525, JP21H01025, JP19H05821 (A.K.).

## Author contributions
T.M. and A.K. proposed the tiling theory. T.D. developed the application to soft matter. A.T. and Y.M. conducted polymer experiments. T.M., A.K., and T.D. wrote the manuscript.

## Competing interests
The authors declare no competing interests.
