## [Peer Review File · Nature Communications]

Aperiodic approximants bridging quasicrystals and modulated structuresReviewer #1 (Remarks to the Author):

The authors published in 2019 a related paper "Metallic-mean quasicrystals as aperiodic approximants of periodic crystals" (ref. 14 in the manuscript). Now the focus on "a series of k-th metallic-mean tilings serving as aperiodic approximants for the honeycomb crystal, while simultaneously redefining this tiling as a metallic-mean IC modulated structure" illustrated in their Fig. 1. The methodology the authors use is more or less the same, however, the focus has been broadened. Of particular interest is the application to soft-matter quasiperiodic systems and approximants, in theory and practice. The methodology is well described and the manuscript is clearly structured and written. I recommend publication of this excellent manuscript as it is.

Reviewer #2 (Remarks to the Author):

In the article "Aperiodic bridging quasicrystals and modulated structures" the authors present a systematic study of hexagonal "metallic-mean" quasicrystals. It refers to 2D structures showing domains of hexagons intersected by boundaries that disturb the periodicity. The structure is based on two characteristic lengths in the system with a ratio $1/\tau_k$, where k is the k'th element of the metallic mean, a sequence of incommensurate ratios.

The article is of great interest for the study of modulated structures, because it illuminates the connection to quasicrystals. The observation of domains of ordered structures separated by boundary regions is frequently found in experiments, especially those with soft matter materials. Very often it is considered that the system is, in principle, crystalline if it would not be disturbed - for practical reasons like a lack of relaxation time. While this may be the fact, but as shown in this article, the modulations of the structure may as well be a systematic quasicrystalline structure, based on the geometry of the basic parts of the system.

For different k, the authors find different quasicrystalline structures with varying properties. Their study involves results of analytic, numerical, and simulational studies, and considers tiling and the higher-dimensional representation of quasicrystals.

A large number of results, like the frequency of vertex types, has been obtained analytically. The findings are compared with results of an experimental study of a polymer blend and a simulation study of a model potential. Both examples are taken from literature, the model potential has been reinvestigated by the authors.

The article is of high value for readers that are working on partially ordered 2D structures, especially of soft matter, and everyone interested in quasicrystals and related topics. However, the article is also informative for other readers, because it is excellently written, and comprehensible. Furthermore, the supplementary gives a helpful introduction into quasicrystal analysis using the example of the given system.

I recommend a publication, if the following minor aspects are considered:

line 78: It might be helpful to refer to Supplementary Note 1

line 109: Please explain or give a reference: How do we get from

"hexagons and parallelograms" to the existence of two sublattices A and B?

Many figures like 2.i, 3.i, 4.d, or satellite peaks in 4.f are too small and hardly readable in the printed version.

Supplementary Note 1: line 43: What do you mean by triple periodicity?

Supplementary Note 12: "We find that three atoms are ... while the others ...center of the L tile."

For me it looks as if in each L tile, there are 6 atoms that are neither in the vicinity of a vertex nor in the center of the L tile.

RESPONSE TO REVIEWERS' COMMENTS

Responses to reviewer #2

We extend our gratitude to the reviewer for his/her meticulous review of the manuscript and for offering insightful comments. Below, we address each of the reviewer's comments individually. The changes are indicated by red letters in the manuscript.

line 78: It might be helpful to refer to Supplementary Note 1

As pointed out by the reviewer, referring to Supplementary Note 1 is indeed helpful for readers. Therefore, we have included the following sentence:

The details of the substitution rules are shown in Supplementary Note 1.

line 109: Please explain or give a reference: How do we get from "hexagons and parallelograms" to the existence of two sublattices A and B?

We appreciate the reviewer for pointing out our careless statement. The hexagonal metallic-mean tiling, composed of even-numbered polygons, is indeed bipartite, meaning its vertices can be divided into two sets (A and B sublattices in this case) such that no two vertices within the same set are adjacent. This property is common in many tilings and graphs, and is fundamental in graph theory and condensed matter physics, such as electronic and magnetic problems. Our inten-

tion here is not to provide rigorous necessary and sufficient conditions for generic bipartite tilings. Therefore, we have simply amended the sentence as follows:

The vertices of the hexagonal metallic-mean tiling can be occupied alternatively by open and solid circles, corresponding to the A and B sublattices, respectively. This property is called bipartite.

Many figures like 2.i, 3.i, 4.d, or satellite peaks in 4.f are too small and hardly readable in the printed version.

Thank you for suggesting improvements to the visibility of the figures. Firstly, we have enlarged Fig. 2i to improve its readability. Next, we believe that the reviewer pointed out the Figs. 3b and d since there is no Fig. 3i. We have enlarged Fig. 3b, and shown only the essential region of the tiling in Fig. 3d to demonstrate the properties of domains. In Fig. 4d, we have replaced the 5th metallic mean tiling with the bronze-mean (3rd) tiling to enhance clarity. Finally, we have enlarged Fig. 4f to improve its readability.

Supplementary Note 1: line 43: What do you mean by triple periodicity?

Thank you for highlighting the terminology issue regarding the use of “triple periodicity”. It’s evident that the meaning was not clear in the previous manuscript. We have removed “triple periodicity” and revised the sentence accordingly:

When the substitution rule is applied to an S tile, one L tile appears. More precisely, the substitution is classified by modulo 3 of k , as illustrated in Supplementary Fig. 1. Specifically, when $k \equiv 1$ and $k \equiv 2 \pmod{3}$, a site represented by an open or filled circle appears at the center, respectively, while no vertex is generated otherwise ($k \equiv 0 \pmod{3}$). Since the arrangement of the resulting L tiles is uniquely determined, we are able to construct the substitution rule for the S tile for any given k . Moreover, we observe the same modulo 3 property for L and P tiles concerning corner sites. Thus, we can extend the substitution rule to encompass any metallic-mean tiling.

Supplementary Note 12: “We find that three atoms are ... while the others ...center of the L tile.” For me it looks as if in each L tile, there are 6 atoms that are neither in the vicinity of a vertex nor in the center of the L tile.

We truly appreciate the reviewer for his/her careful reading. As pointed out by the reviewer, the description of six atoms was lacking in the previous manuscript. We have amended the corresponding sentence as follows:

We find that three atoms are located at the triangle vertices on each vertex. In addition, within each L tile, three atoms are positioned at the triangle vertices at the center, while six atoms are located in the vicinity of each of the six edge centers.

To fit the Nature Communication formatting instructions, we have furthermore modified our manuscript. We hope that our manuscript is now suitable for publication.

Changes to fit the Nature Communication

In the main text, we have

- added headings “INTRODUCTION” and “RESULTS”
- added asterisk and dagger in page 1 and emails in page 11 for corresponding authors
- modified the abstract (150 words or fewer)
- added final page number for Ref. 3
- removed “novel” in the final sentence
- introduced “transmission electron microscopy (TEM)”, “fast Fourier transform (FFT)”, and “Onoda-Steinhardt-DiVincenzo-Socolar (OSDS)” to avoid the abbreviations without definitions
- removed comma just after “a”, “b”, ... in the caption of figures

In the Supplementary information, we have

- replaced “upper” and “lower” into “a” and “b” in the Supplementary Fig. 4
- replaced the term “Equation”, which refers to an equation in the Supplementary Information, with “Supplementary Equation”
- corrected the citation style

Reviewer #1 (Remarks to the Author):

As stated in my previous review, I support the publication of the manuscript as it is. This applies even more to the slightly revised version of the manuscript.

Reviewer #2 (Remarks to the Author):

As mentioned in the last referee report, the article is very well suited for the journal. The authors have improved now the text and the figures as suggested by my last referee report.

In this form, I recommend the publication of the article